# Graphene-Delivered Insecticides against Cotton Bollworm

**DOI:** 10.3390/nano12162731

**Published:** 2022-08-09

**Authors:** Zhiwen Chen, Jianguo Zhao, Zehui Liu, Xiuli Bai, Weijia Li, Zhifang Guan, Ming Zhou, Hongwei Zhu

**Affiliations:** 1Hainan Yazhou Bay Seed Laboratory, Sanya 572025, China; 2National Key Laboratory of Plant Molecular Genetics and National Center for Plant Gene Research, Institute of Plant Physiology and Ecology/CAS Center for Excellence in Molecular Plant Sciences, Chinese Academy of Sciences, Shanghai 200032, China; 3Institute of Carbon Materials Science, School of Chemistry and Chemical Engineering, Shanxi Datong University, Datong 037009, China; 4School of Mechanical and Transportation Engineering, Guangxi University of Science and Technology, Liuzhou 545006, China; 5State Key Laboratory of New Ceramics and Fine Processing, School of Materials Science and Engineering, Tsinghua University, Beijing 100084, China

**Keywords:** graphene, insecticide, lambda-cyhalothrin, cyfluthrin, synergistic mechanism

## Abstract

Nanopesticides can facilitate controlled release kinetics and efficiently enhance the permeability of active ingredients to reduce the dosage and loss of pesticides. To clarify the synergistic mechanism of graphene–insecticide nanocarriers against cotton bollworm, treatment groups, namely, control, graphene (G), insecticide (lambda-cyhalothrin (Cyh) and cyfluthrin (Cyf)), and graphene-delivered insecticide groups were used to treat the third-instar larvae of cotton bollworm. The variations in phenotypes, namely, the body length, body weight, and mortality of the cotton bollworm, were analyzed. The results show that graphene enhances the insecticidal activity of lambda-cyhalothrin and cyfluthrin against cotton bollworm. The two graphene-delivered insecticides with optimal compositions (3:1) had the strongest inhibitory effects and the highest mortality rates, with the fatality rates for the 3/1 Cyh/G and Cyf/G mixture compositions being 62.91% and 38.89%, respectively. In addition, the 100 μg/mL Cyh/G mixture had the greatest inhibitory effect on cotton bollworm, and it decreased the body length by 1.40 mm, decreased the weight by 1.88 mg, and had a mortality rate of up to 61.85%. The 100 and 150 μg/mL Cyh/G mixtures achieved the same mortality rate as that of lambda-cyhalothrin, thus reducing the use of the insecticide by one-quarter. The graphene-delivered insecticides could effectively destroy the epicuticle spine cells of the cotton bollworm by increasing the permeability and, thus, the toxicity of the insecticides.

## 1. Introduction

An efficient and sustainable agricultural and forestry system is crucial to human survival [1]. The global production of 3 billion tons of food annually requires 4 million tons of pesticides [2]. Pesticides are commonly used in the agricultural sector to prevent and manage agricultural pests for agro-forestry plant protection [3]. However, the resistance of target organisms and the large amount of pesticide residues lower the efficiency of target insect control, and the intensification and repeated use of insecticides are becoming increasingly restricted [4,5]. In addition, current agricultural and forestry production is facing serious environmental degradation [6,7], and environmental damage from insecticide overuse is a major concern when attempting to meet human demands for food and high-quality wood [8].

Nanomaterials have large specific surface areas and have been widely used in physics, electrochemistry, medicine, and agriculture [9,10]. The development and potential applications of plant protection products called “nanopesticides” have attracted great attention [11,12,13]. Nanopesticides can facilitate controlled release kinetics and efficiently enhance the permeability of active ingredients to reduce the dosage and loss of pesticides [14]. Among the nanopesticides, graphene has attracted extensive attention with its improved efficiency of pesticide utilization [15]. It has been reported that 40% of pesticide residues remain in leaves with the use of graphene coated with copper selenide, with accurate controlled release, and the mortality rate of powdery butterfly larvae can be increased to more than 35% [16]. Tong et al. revealed that polydopamine-coated graphene, as a pesticide carrier, can improve the utilization efficiency and reduce the loss of water-soluble pesticides caused by rain or irrigation water [17]. Wang et al. showed that graphene mixed with three types of pesticides, namely, pyridaben, chlorpyrifos, and beta-cyfluthrin, can enhance the activity of these pesticides against spider mites [18]. Subsequently, they demonstrated the excellent synergistic activities of graphene with three types of insecticides, namely, β-cyfluthrin, monosultap, and imidacloprid, against the Asian corn borer [19]. All these studies found that graphene can improve the efficiency of pesticide delivery and enhance insecticidal efficacy.

Lambda-cyhalothrin (Cyh) and cyfluthrin (Cyf) are common insecticides used to control a wide variety of agroforestry pests worldwide [20,21,22]. In this study, Cyh and Cyf were mixed with graphene to study the synergistic insecticidal mechanism against cotton bollworm (*Helicoverpa armigera*), one of the most destructive pests worldwide. Third-instar cotton bollworm larvae were treated with distilled water (defined as the CK group), graphene, two insecticides, or graphene–insecticide mixtures. By comparing the body length, weight, and mortality of the cotton bollworm, the optimal ratio of the graphene and insecticide mixture was examined. The synergistic insecticidal mechanisms of the graphene–insecticide mixtures were analyzed. The results show that graphene-based nanoinsecticides may have broad application prospects in the plant protection field.

## 2. Results and Discussion

### 2.1. Workflow of the Synergistic Mechanism of Graphene-Delivered Insecticide

Distilled water (CK), graphene, insecticides, and graphene-delivered insecticides were used to treat the third-instar larvae of cotton bollworm for 24 h. The changes in the indices of the body length, body weight, and mortality of the cotton bollworm were measured. No acute toxicity of graphene was found compared with the distilled water. However, compared to the insecticides, enhanced insecticidal activities were observed in the graphene–insecticide group, whose synergistic mechanism was related to the damage caused to the epicuticle spine cells of the cotton bollworm (Figure 1A).

### 2.2. Characterizations of Graphene, Insecticides, and Graphene–Insecticide Mixtures

Graphene, insecticides (Cyh and Cyf), and graphene–insecticide mixtures were analyzed using SEM. As shown in Figure 1B,C, the graphene sheets were thin, transparent, and smooth, with a slightly wrinkled and undulated surface structure. Both Cyh and Cyf were absorbed on the surface of graphene with granular structures (Figure 1D–G).

### 2.3. Characterization of Surface Morphology of Graphene

The SEM and TEM images further reflect the surface morphology and internal structure of the graphene, respectively. In Figure 2A,B, the stratification of graphene is obvious, and the spacing between layers gradually becomes larger. The SEM results also showed that the graphene used in this study had a fold or transparent morphology. The TEM images showed that the thickness of the graphene sheet was about 1~3 nm, indicating that the number of graphene layers was less than 5 (Figure 2C,D).

### 2.4. Composition and Structure Analyses of Graphene

The graphene sheets were analyzed using an infrared spectrometer to obtain an infrared spectrum (Figure 3A). The results show that there were stretching vibrations of the hydroxyl group (–OH) at 3421 cm^−1^ and 3207 cm^−1^, which were derived from H_2_O and carboxyl groups (–COOH), respectively. The stretching vibration peak of C–O appeared at 1122 cm^−1^. The double peaks of 2919 cm^−1^ and 2848 cm^−1^ were caused by the stretching vibration of the –CH2– group, while the stretching vibration absorption peak of carbonyl C=O appeared at 1720 cm^−1^. At 1637 cm^−1^, there was a stretching vibration region of carbon–carbon double bonds (C=C). The out-of-plane bending vibration of C–H was observed at 655 cm^−1^. These results indicate that graphene had oxygen-containing groups, such as –COOH and –OH, on its edge, which can form hydrogen bonds with water. Raman spectroscopy is another method most commonly used to characterize graphene. Figure 3B shows that there are three obvious peaks in the Raman spectrum: the D peak (1344 cm^−1^), the G peak (1575 cm^−1^), and the 2D peak (2682 cm^−1^). The intensity ratio of I_D_/I_G_ = 0.625 indicated that the defect density of the graphene was low. The intensity ratio of the G peak to the 2D peak was greater than 1, indicating that the graphene used in this experiment had multilayer sheets, which is consistent with the TEM results. The particle size analysis diagram shows the distribution of the graphene sheet size (Figure 3C). It can be seen in Figure 3C that the graphene sheets ranged from 10 nm to 88 nm, and the sheet diameters of D10 (cumulative distribution = 10%), D50 (cumulative distribution = 50%), and D90 (cumulative distribution = 90%) were 12 nm, 20 nm, and 37 nm, respectively.

### 2.5. Insecticidal Activity of Graphene–Insecticide Mixtures against Cotton Bollworm

According to the optimal concentrations (100 μg/mL) of Cyh and Cyf for cotton bollworm control [23], seven Cyh/G and Cyf/G mixtures were prepared with different compositions (1/4, 1/3, 1/2, 1/1, 2/1, 3/1, and 4/1), and their total concentrations were 100 μg/mL. Distilled water treatment was used for the control group. Fourteen graphene–insecticide mixtures and distilled water were used to treat third-instar *H. armigera* for 24 h, and the insecticidal activities against *H. armigera* were evaluated, namely, the changes in the body length, body weight, and mortality of *H. armigera* larvae.

The results of the treatments with Cyh/G mixtures are shown in Figure 4. Compared with the control group, the 2/1, 3/1, and 4/1 compositions significantly inhibited the growth and development of the cotton bollworm, while the 1/4, 1/3, 1/2, and 1/1 Cyh/G mixtures slightly inhibited the growth and development of the cotton bollworm (Figure 4A). In the control group, the body length increased by 1.34 mm, while in the Cyh/G-treated groups, it decreased by 0.08~1.39 mm, among which the 3/1 composition resulted in a sharp decrease of 1.39 mm (Figure 4B). In the control group, the weight increased by 13.28 mg, while in the groups treated with Cyh/G mixtures, it decreased by 0.70~6.17 mg, among which the 3/1 composition showed a sharp decrease of 6.17 mg (Figure 4C). The mortality of the control group was 6.18%, and the fatality rates of the Cyh/G mixture groups ranged from 11.73% to 62.91%, among which the fatality rates for the 3/1 and 4/1 compositions were 62.91% and 58.50%, respectively, significantly higher than those of the other groups (Figure 4D). The average fatality rate under the 3/1 composition was higher than that of the 4/1 composition, but there was no significant difference between the two groups (Figure 4D). The above results show that the 3/1 Cyh/G mixture had the most obvious inhibitory effect on the growth and development of the cotton bollworm, as well as the highest mortality rate. Therefore, the 3/1 Cyh/G mixture was selected as the optimal composition for use in subsequent studies.

The results of the treatments with the Cyf/G mixtures are shown in Figure 5. Compared with the control group, the 2/1, 3/1, and 4/1 compositions significantly inhibited the growth and development of the cotton bollworm, while the 1/4, 1/3, 1/2, and 1/1 Cyf/G mixtures slightly inhibited the growth and development of the cotton bollworm (Figure 5A). The body length of *H. armigera* in the control group increased by 2.20 mm, while the body length of those treated with Cyf/G mixtures changed by −0.19~1.82 mm, among which the 3/1 composition caused a decrease of 0.19 mm (Figure 5B). The weight increase in the control group was 15.59 mg, and the weight increase in the Cyf/G mixture groups was 1.99~10.25 mg, among which the 3/1 composition insecticide caused an increase of 1.99 mg, which was significantly lower than that of the other groups (Figure 5C). The mortality of the control group was 8.33%, and the fatality rates of the Cyf/G mixtures ranged from 16.67% to 38.89%, among which those of the 3/1 and 4/1 compositions were 38.89% and 33.33%, respectively, significantly higher than those of the other groups (Figure 5D). The average fatality rate of the 3/1 composition was higher than that of the 4/1 composition, but there was no significant difference between the two groups (Figure 5D). The above results show that the Cyf/G mixture (3/1) had the most obvious inhibitory effect on the growth and development of the cotton bollworm, as well as the highest mortality rate. This result is consistent with that of Cyh/G, with the optimal ratio of 3/1, but the toxicity of Cyf/G is lower than that of Cyh/G.

### 2.6. Insecticidal Activity of Graphene, Insecticides, and Graphene–Insecticide Mixtures

In the subsequent experiments, four concentrations (37.5, 75, 100, and 150 μg/mL) of graphene, Cyh, Cyf, Cyh/G, and Cyf/G mixtures (3/1) were prepared and used to treat third-instar *H. armigera*. Their insecticidal activities against *H. armigera* were evaluated after 24 h of treatment, namely, the changes in the body length, weight, and mortality of the larvae.

As shown in Figure 6, compared with the control group, the treatments with different concentrations of graphene had no effect on the growth and development of *H. armigera*. The Cyh and Cyh/G mixtures (3/1) significantly inhibited the growth and development of *H. armigera*, and the higher the concentration, the more significant the inhibitory effect (when below 100 μg/mL concentrations) (Figure 6A).

The body length of *H. armigera* in the control group increased by 1.74 mm, while the body length of those in the graphene groups increased by 1.56~2.94 mm. In contrast, the body length of *H. armigera* in the Cyh group decreased by 0.13~1.06 mm, and the body length of those in the Cyh/G mixture group decreased by 0.22~1.40 mm. The Cyh/G mixture (100 μg/mL) showed the optimal effect, causing a 1.40 mm decrease in body length (Figure 6B). The body weight of the control group increased by 20.33 mg, while for the graphene group, it increased by 16.91~21.91 mg. The weight change in the Cyh group was −1.76~3.44 mg, and that in the Cyh/G mixture group was −1.88~2.02 mg. Likewise, the body weight of the Cyh/G group (100 μg/mL) decreased the most, with a change of 1.88 mg (Figure 6C).

After treatment, the mortality rate was 0.83% in the control group, 1.82~5.30% in the graphene group, 22.00~63.67% in the Cyh group, and 15.45~62.46% in the Cyh/G group (Figure 6D). At 100 and 150 μg/mL, the lethality rates of Cyh in the cotton bollworm were 61.85% and 63.67%, respectively, while the lethality rates of the Cyh/G mixture in the cotton bollworm were 61.33% and 62.46%, respectively. There was no significant difference in the mortality rates among these four groups. The above results show that graphene caused no contact toxicity to *H. armigera* and did not inhibit the growth or development of *H. armigera*, and its mortality rate was not significantly higher than that of the control group. The body length and weight of *H. armigera* were significantly inhibited by the Cyh and Cyh/G mixtures at the optimal lethal concentration of 100 μg/mL. Thus, using the Cyh/G mixture can reduce the use of Cyh by one-quarter.

The results of the treatment with Cyf are shown in Figure 7. Compared with the control group, the graphene treatment had no effect on the growth and development of the cotton bollworm, while both Cyf and Cyf/G significantly inhibited the growth and development of the cotton bollworm (Figure 7). The body length changed by 5.45 mm in the control group, 3.50~5.18 mm in the graphene group, −0.47~2.07 mm in the Cyf group, and 0.12~0.98 mm in the Cyf/G group. The body length of the Cyf group (150 μg/mL) increased the least (−0.47 mm) (Figure 7A). The body weight changed by 24.12 mg in the control group, by 12.21~23.14 mg in the graphene group, by −0.04~8.05 mg in the Cyf group, and by 3.01~10.91 mg in the Cyf/G group. Similarly, the 150 μg/mL Cyf group had the lowest weight gain (−0.04 mg) (Figure 7B). Cyf alone was superior to the Cyf/G mixture in terms of limiting increases in body length and weight.

Compared with the control group (mortality: 2.78%), the mortality of the cotton bollworm in the graphene group ranged from 2.08% to 11.17%. Cyf caused high mortality, ranging from 14.96% to 41.67%, and the Cyf/G mixture led to a higher fatality rate with the four concentrations, ranging from 18.75% to 45.08% (Figure 7C). At the high concentrations of 100 and 150 μg/mL, the mortality rates of Cyf in the cotton bollworm were 20.83% and 41.67%, respectively, while the mortality rates of Cyf/G were 37.50% and 45.08%, respectively, both higher than that of Cyf alone (Figure 7C). The results show that the toxicity of graphene to the cotton bollworm became weak after 24 h of treatment, and Cyf could significantly inhibit the length and weight gain of the cotton bollworm at high concentrations (>75 μg/mL). The Cyf/G mixture was less effective than Cyf alone in inhibiting the length and weight gain of the cotton bollworm. However, both Cyf/G mixtures (100 and 150 μg/mL) had higher mortalities, thus enabling the reduced use of Cyf.

### 2.7. Graphene–Cyh Mixture Effectively Destroys the Epicuticular Spines of Cotton Bollworm

To analyze the mechanism of the high insecticidal activity of Cyh/G against cotton bollworm, the changes in the epicuticular cell structure of the cotton bollworm treated with different concentrations of graphene, Cyh, and Cyh/G were observed using SEM (Figure 8). In the control group, the morphology of the epicuticle of the cotton bollworm treated with distilled water was arranged in a regular and smooth manner, and the structure of the cuticular layer was intact (Figure 8A). The cotton bollworm epicuticles treated with graphene were arranged in an irregular folding pattern; the structure of the epicuticular spines was slightly damaged. The higher the concentration of graphene, the more obvious the damage (Figure 8B). The epicuticular morphology of the cotton bollworm treated with Cyh also showed regular folding, and the structure of the epicuticular spines was not damaged, not even at the high concentrations of 100 and 150 μg/mL (Cyh-100 and Cyh-150, respectively) (Figure 8C). In contrast, the morphological integrity of the epicuticle of the cotton bollworm treated with Cyh/G was worse, and the structure of the epicuticle was seriously damaged (Figure 8D). At concentrations of 75 μg/mL (Cyh/G-75), 100 μg/mL (Cyh/G-100), and 150 μg/mL (Cyh/G-150), the spines and the cement layer structure of the epicuticle were seriously damaged, and the degree of damage was the highest at 100 and 150 μg/mL (Cyh/G-100 and Cyh/G-150, respectively) (Figure 8D). The above results indicate that graphene can cut and destroy the epicuticle of cotton bollworm, which increases the contact toxicity of Cyh and leads to insect death, thus achieving the purpose of high insecticidal activity.

## 3. Potential Application and Prospects

Nanomaterials (with diameters of less than 100 nm) can cross plant biological barriers (such as cell walls) and enter vascular bundle tissue, providing a new path for the delivery of pesticides [24,25], and they can also be used as sensors to examine plant status at any time in order to enhance the ability of plants to cope with environmental stress, thus improving yields in agriculture and forestry [26,27,28,29]. These materials show promise in scientific and technological revolutions in agricultural and forestry fields [30].

Nanopesticides or nano-plant protection products represent a hopeful scientific development that offers a variety of benefits, including increased effectiveness, durability, and a reduction in the amount of active ingredients (AIs) that are used to protect crops against diseases, insects, and weeds [31,32]. Successful examples of nanopesticides against major agricultural insect pests exist. For instance, researchers have finished examining the controlled-release formulations of β-cyfluthrin developed using poly(ethylene glycol) (PEG)-based amphiphilic copolymers, and the rate of release of encapsulated β-cyfluthrin from the nano-formulations was reduced by increasing the molecular weight of PEG [33]. Their results indicated that the developed β-cyfluthrin nano-formulation had prolonged activity and was more effective against *Callosobruchus maculatus* than a commercial formulation [33]. Nano-acephate (80~120 nm), prepared by the encapsulation of acephate in PEG, was found to be a more promising against *Spodoptera litura* and did not induce any cytotoxicity in a human fibroblast cell line [34].

In addition, chitosan, a biodegradable polymer obtained from the deacetylation of chitin, has attracted considerable interest for its role in achieving the effective and controlled release of pesticide activity. Nanostructured pyrifluquinazon prepared using chitosan as a carrier showed the best lethal efficiency against the green peach aphid *Myzus persicae* after a 14-day treatment [35]. Recently, neem-based nano-capsules prepared using biodegradable polymers poly-ε-caprolactone (PCL) and poly-β-hydroxybutyrate (PHB) caused a high mortality rate in *Bemisia tabaci*, a serious pest of many crops [36].

Graphene has been widely used in nano-plant protection fields. Previous studies showed that graphene has multifunctional synergistic activity toward insecticides against spider mites and Asian corn borers [18,19]. In these studies, firstly, graphene was mixed with three types of pesticides (pyridaben (Pyr), chlorpyrifos (Chl), and beta-cyfluthrin (Cyf)) against spider mites [18]. The results demonstrated that graphene enhanced the toxicity of the three types of pesticides. Compared to the pesticides alone, the graphene–Cyf, graphene–Pyr, and graphene–Chl mixtures exhibited 1.5–1.8-fold higher mortality rates against spider mites. In a subsequent study, the same authors investigated the synergistic activity of graphene mixed with insecticides β-cyfluthrin (Cyf), monosultap (Mon), and imidacloprid (Imi) against the lepidoptera insect Asian corn borer (ACB) [19]. The graphene–insecticide mixtures exhibited 1.8–2.1-fold higher contact toxicities than the individual insecticides at all concentrations used. The previous studies found that graphene increased the insecticide’s insecticidal efficiency regardless of the concentration, and, in our study, the graphene–Cyf/Cyh mixtures increased the mortality rate of the insecticide against cotton bollworm at concentrations of 100 and 150 μg/L, but no synergistic effect was observed at low concentrations. This inconsistency may be caused by the difference in the way the graphene is mixed with the pesticides. However, graphene can indeed be a carrier of pesticides adsorbed on the surface of insects and disrupt the cement layer cell of insects to improve the dispersibility and utilization efficiency of insecticides. Graphene can also make the Asian corn borer “fatter” but shorten its life span [37]. Our study reveals that graphene has synergistic activity in concert with Cyh and Cyf insecticides against cotton bollworm. The differences in the synergistic mechanism of the insecticides versus the graphene–insecticide mixtures are that the mixtures can effectively destroy the epicuticular spine cells of cotton bollworm, thus providing a new channel for insecticide penetration into the cuticle cell and increasing the toxicity of insecticides.

## 4. Materials and Methods

### 4.1. Preparation and Characterization of Graphene and Graphene–Insecticide Mixtures

Graphene materials were prepared by the Institute of Carbon Materials Science, Shanxi Datong University [38,39]. The graphene–insecticide mixtures were prepared with a mass ratio of 3:1 with distilled water. The mixtures were treated hyperacoustically at 100 Hz for 30 min, frozen, stored in a −80 °C refrigerator, and then dried in a freeze-dryer (Beijing Boyikang Experimental Instrument Co., Ltd. (Beijing, China)Freeze-dryer FD-1A-50) for 72 h. Then, the dried samples were characterized using scanning electron microscopy (SEM, TESCAN MAIA 3 LMH, Brno, Czech Republic). Cyh/Cyf (7.50 mg) and graphene (2.50 mg) were directly mixed and dissolved in 10 mL distilled water, and then these two solutions were diluted to a series of concentrations with distilled water for subsequent experiments.

### 4.2. Acquisition of Insecticides and Cotton Bollworm

Two insecticides (Cyh and Cyf) were purchased from the Zhongbao Green Agriculture Group, the Institute of Plant Protection, the Chinese Academy of Agricultural Sciences, and the second-instar larvae of cotton bollworms were purchased from KEYUN Biology Co., Ltd. (Beijing, China).

### 4.3. Cotton Bollworm Culture and Insecticide Spraying

The second-instar larvae of the cotton bollworms were put into a 24-cell culture box with 1 cm^3^ fodder in each cell for one day. Then, they were reared in a chamber at 28 °C under a relative humidity of 40% and a photoperiod of 14 h of darkness and 10 h of light, with a light intensity of 33%. Two days later, the third-instar larvae of the cotton bollworms were treated with either 1.5 mL of distilled water (CK), graphene, insecticides (Cyh and Cyf), or graphene–insecticide mixtures (Cyh/G and Cyf/G), at concentrations of 37.5, 75, 100, and 150 μg/mL. There were four replicates for each group, with 12 cotton bollworms in each treatment. After treatment, the cotton bollworms were reared under the above conditions again. The statistical indices of the body length, weight, and mortality of the treated cotton bollworms were measured at 0 h and 24 h. A tested cotton bollworm was considered dead if it could not wriggle when prodded with an insect needle.

### 4.4. Identification of Morphological Changes in Cotton Bollworm Epicuticle Cells

The morphology of cotton bollworm epicuticle cells was observed using SEM to illustrate the dynamic changes in the ultrastructure of the epicuticle cells. The third-instar larvae were treated with distilled water, graphene, Cyh, and Cyh/G. Then, the cotton bollworm larvae were fixed with 2.5% glutaraldehyde with a vacuum pump in an ice bath for 30 min, followed by 4 h of incubation at 4 °C and three washes with phosphate-buffered saline (PBS). After that, the larvae were postfixed 3 times with phosphoric acid buffer (PB) (pH 7.4, 0.1 M) for 15 min. The PB was fixed with 1% osmium at room temperature protected from light for 1~2 h, and then it was used to rinse the samples 3 times for 15 min. Subsequently, the larvae were dehydrated with an ethanol series (30–50-70–80-90–95-100–100% alcohol for 15 min each time and isoamyl acetate for 15 min) and put into a critical point drying instrument to dry. Finally, the samples were subjected to conductive processing and examined under SEM.

## 5. Conclusions

Two insecticides were mixed with graphene to treat the third-instar larvae of cotton bollworm. The effects on the body length, body weight, and mortality of the cotton bollworm were analyzed among control, graphene, insecticide (Cyh and Cyf), and graphene-delivered insecticide groups. The results show that graphene–insecticide mixtures (mass composition: 3:1) greatly enhanced the insecticidal activities of Cyh and Cyf against cotton bollworm. The 3/1 Cyh/G treatment had the greatest inhibitory effect on cotton bollworm, with a 1.39 mm decrease in body length, a 6.17 mg decrease in weight, and a 62.91% mortality rate. In contrast, in the cotton bollworm, the 3/1 Cyf/G treatment led to a 0.19 mm decrease in body length, a 1.99 mg decrease in weight, and a 38.89% mortality rate. Compared to the use of the insecticides alone, the graphene–insecticide mixtures significantly inhibited the body length and weight gain of the cotton bollworm and achieved higher mortalities. The 100 μg/mL Cyh/G mixture decreased the body length by 1.40 mm, decreased the weight by 1.88 mg, and had a 61.85% mortality rate. In contrast, the 150 μg/mL Cyf/G mixture had a 45.08% mortality rate, which is higher than that of 100 μg/mL. The graphene-delivered Cyh could effectively destroy the epicuticle and spine cells of the cotton bollworm. This result proves that graphene has synergistic activity with insecticides, and this might have big implications in the plant protection field.

## Figures and Tables

**Figure 1 nanomaterials-12-02731-f001:**
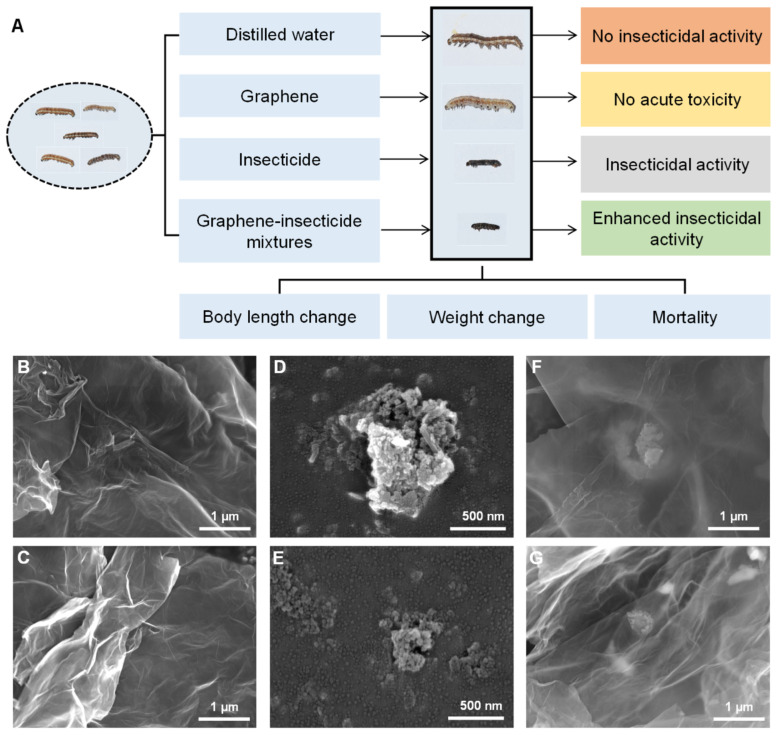
(**A**) Workflow of the synergistic mechanism study of graphene-delivered insecticide. Characterizations of graphene, insecticides, and formulated graphene–insecticide mixtures: SEM images of (**B**,**C**) graphene, (**D**) Cyh, (**E**) Cyf, (**F**) Cyh/G (3/1), and (**G**) Cyf/G (3/1).

**Figure 2 nanomaterials-12-02731-f002:**
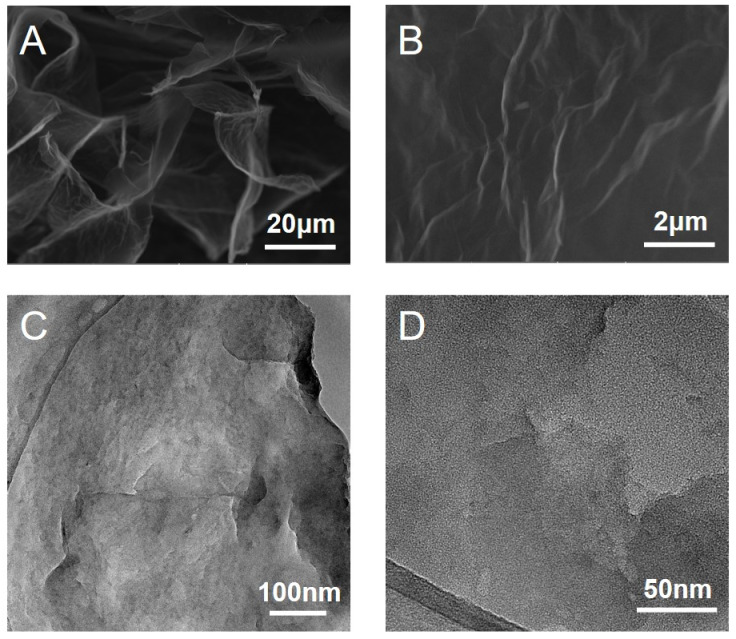
Characterizations of the surface morphology of graphene. (**A**,**B**) SEM images of graphene and (**C**,**D**) TEM images of graphene.

**Figure 3 nanomaterials-12-02731-f003:**
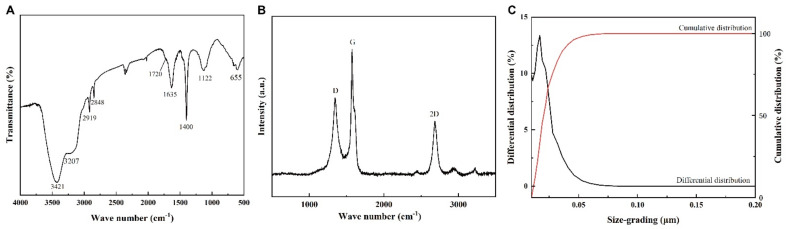
Composition and structure analyses of graphene. (**A**) FT-IR spectra of graphene, (**B**) Raman spectra of graphene, and (**C**) size distribution of graphene.

**Figure 4 nanomaterials-12-02731-f004:**
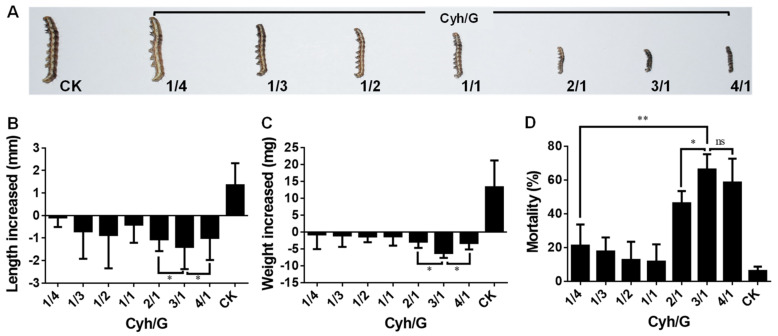
Insecticidal activities of Cyh/G mixtures against cotton bollworm. (**A**) Phenotypic effects. (**B**) Body length change, (**C**) weight change, and (**D**) mortality of the cotton bollworm. CK represents distilled water control. The total concentration of all treatment groups was 100 μg/mL. Each treatment was performed with four biological replications. Note: statistically significant differences (α = 0.05 and 0.01 levels) of values are indicated with “*” and “**”, respectively. “ns”: not significant.

**Figure 5 nanomaterials-12-02731-f005:**
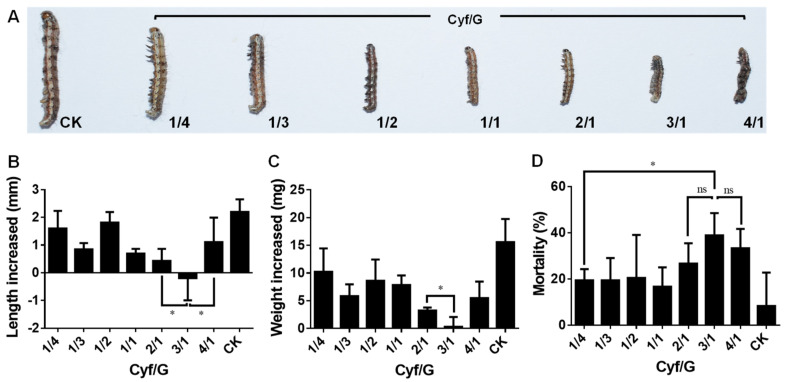
Insecticidal activity of Cyh/G mixtures against cotton bollworm. (**A**) Phenotypic effects. (**B**) Body length change, (**C**) weight change, and (**D**) mortality of the cotton bollworm. CK represents distilled water control. The total concentration of all treatment groups was 100 μg/mL. Each treatment was performed with four biological replications. Note: statistically significant differences (α = 0.05 level) of values are indicated with “*”. “ns”: not significant.

**Figure 6 nanomaterials-12-02731-f006:**
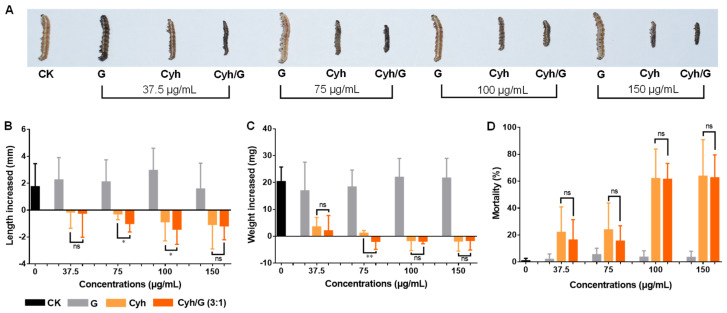
Insecticidal activity of different concentrations of graphene, Cyh, and Cyh/G against cotton bollworm. (**A**) Phenotypic effects. (**B**) Body length change, (**C**) weight change, and (**D**) mortality of the cotton bollworm. CK represents distilled water control. Note: statistically significant differences (α = 0.05 and 0.01 levels) of values are indicated with “*” and “**”, respectively. “ns”: not significant.

**Figure 7 nanomaterials-12-02731-f007:**
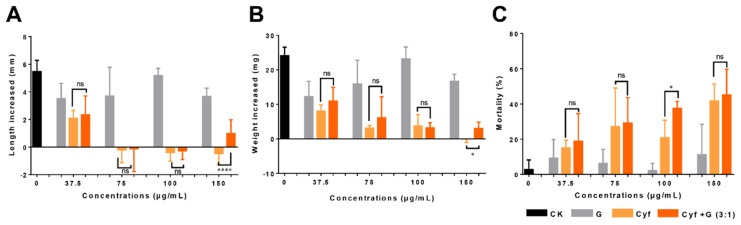
Insecticidal activity of different concentrations of graphene, Cyf, and Cyf/G against cotton bollworm. (**A**) Body length change, (**B**) weight change, and (**C**) mortality of the cotton bollworm. Note: statistically significant differences (α = 0.05 and 0.0001 levels) of values are indicated with “*” and “****”, respectively. “ns”: not significant.

**Figure 8 nanomaterials-12-02731-f008:**
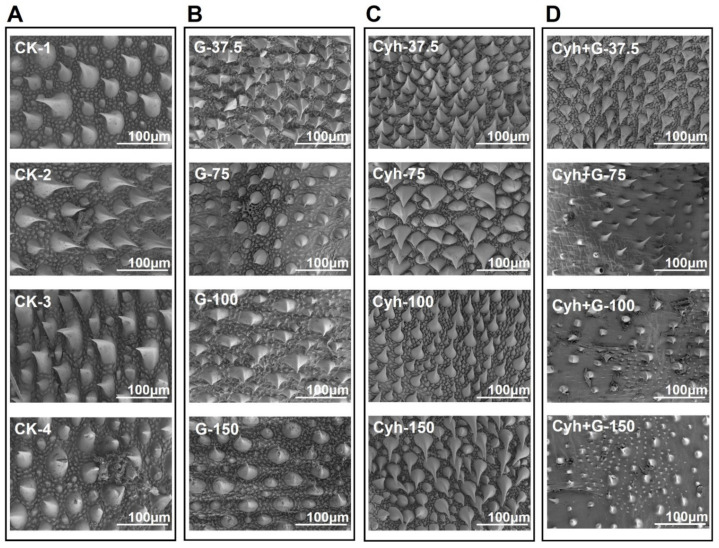
SEM images of the cotton bollworm epicuticle. (**A**) Control group. (**B**) Treated with graphene. (**C**) Treated with Cyh. (**D**) Treated with Cyh/G. CK represents distilled water control.

## Data Availability

Not applicable.

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
