# Peer review of "Graphene-Delivered Insecticides against Cotton Bollworm"

_nanomaterials, 2022, doi:10.3390/nano12162731_

Round 1
Reviewer 1 Report
Nanomaterials-1808932 reports a study of the effects of graphene on insecticide activity against the cotton bollworm for two commonly used insecticides. The results are likely to be of interest to scientists investigating the potential for nanopesticides to improve the efficiency of existing pesticides by improving uptake and/ or reducing amount of pesticide needed. This is currently an area of considerable scientific interest and one with potential for introduction of more sustainable agricultural practices. However, the paper lacks clarity and details in a number of areas, as summarized below, in order to be publishable.
1. The graphene references in the sentence starting on line 47 are mostly for applications other than nanopesticides. It would be clearer to separate general applications of graphene from nanopesticide applications.
2. Figure 1 and Figure 2, as well as the text should note that the distilled water control is labeled as CK. This defined in the Materials and Methods but since that is at the end of the paper, it is not immediately obvious how to interpret the initial figures, especially since CK is not a standard acronym to use for a distilled water control.
3. The Figure 1 caption should provide the graphene/insecticide ratio used for panels F and G..
4. The captions for Figures 2 and 3 should note the concentration used for ease of comparison to later figures and it would also be useful to note the number of replicates. It is worth noting that the error bars for many data points in Figure 2 are larger than the amount of change; the authors should consider whether the number of significant figures given for the data that discusses this figure are appropriate. Finally, line 86 notes that 100 µg/mL is the optimal concentration for both Cyh and Cyf, so that is what is used for these experiments. A reference is needed and it should be noted if this is for cotton bollworm larvae as investigated here
5. Lines 131-132 notes the lower toxicity of Cyf/G vs Cyh/G and later the same effect is evident for the insecticides alone. Has this been observed previously? What is the difference in structure and what is known about possible differences in activity/mechanism for the two insecticides?
6. Line 138, gradients should be “concentrations”.
7. Lines 142-145 note that higher concentrations give a more significant inhibitory effect (Figure 4). This is true at low concentrations, but 100 µg and 150 µg give more or less identical results, so this statement should be clarified.
8. The combined results show that there are weight, length and mortality changes induced by the various insecticide and graphene-insecticide mixtures that are tested and that Cyh and Cyf do not behave in exactly the same manner. What is not clear (to a non-plant expert) is what the most desirable result is. I would assume that one wants to kill as many of the cotton bollworm larvae as possible, so the variable results for height and length are of less interest. More discussion of this is needed.
9. For the SEM data in Figure 6, it seems odd that 100 µg Cyh has a much larger effect than either 75 µg or 150 µg. Is this a reproducible effect and do the authors have an explanation?
10. Potential applications.
a) Lines 232-233 note that a previous nano-formulation of cyfluthrin has been studied. Please provide details on the type of nano-formulation (carrier?) and the mode of action, if reported.
b) Lines 244-253 describe the previous work on effects of graphene on insecticide activity. It would be useful to provide more detail on the mechanism and the quantitative changes observed for comparison with the present study.
c) Lines 250-253 claim that the graphene insecticide mixtures can destroy the epicuticular spine cells, providing a new channel for insecticide penetration. However, based on Figure 6, one can also see a similar effect for Cyh at 100 µg. This might indicate that the pesticide alone has some capacity to induce the same damage as the Cyh-graphene mixture.
11. Both the abstract and conclusions describe only some of the results and do not capture the complexity associated with changes in the 3 parameters measured as a function of insecticide and concentration. A more detailed summary is needed
12. The conclusions claim (lines 263-265) that the present results have great implications for plant protection. In that context it would be important to comment on whether the use of graphene-insecticide mixtures is cost effective, by comparison to the insecticide alone. It would also be relevant to question whether the improvement observed for the mixtures is large enough to make them an attractive alternative to insecticide alone.
13. The Materials and Methods notes that the graphene insecticide mixtures are dried. However, it appears that the larvae are treated with solutions, so presumably the mixtures are redispersed. Please add details.
Reviewer 2 Report
The topic of the article is clear and timely, well organized with proper characterizations and data. Additionally, some of the characterizations are needed for the betterment of manuscript. Therefore, I suggest that this paper can be accepted after minor revisions.
1) The language of the current version should be further refined. Especially from 186-195.
2) In characterizations, only SEM imaging was shown as proof. We recommend authors to produce IR spectroscopy results for the better understanding.
3) The weight loss of the graphene insecticides can also be confirmed through TGA curve by comparing all other prepared composites which will further improve the results and discussions part. So, we recommend authors to carry out the experiments and discuss the results.
4) The role of Graphene is not clearly discussed.
Reviewer 3 Report
The manuscript reports quite interesting results on using graphene to facilitate the effects of certain insecticides (lambda-cyhalothrin and cyfluthrin). However, there are some issues which make this paper unsuitable for publication:
- the mechanism of the insecticidal effect is unclear. The authors claim that the the composites (Graphene-Cyh mixture) effectively damage the epicuticular spines of the cotton bollworms. But they provide no information on why this happens. Moreover, no other tests were performed. The authors simply show some SEM of apparently damaged cuticle, however it is unclear if any material is being taken up/congested by the worms. In this sense, the study is descriptive and provides no insights on mechanisms or applicability
- the characterisation of the material was not performed. They simply show SEM of some aggregates. No loading efficiency, no release profiles, no colloid stability, nothing. "Nanomaterials" is a journal which supposed to provide insights into materials structure, properties and functions. Here none is shown.
Technical issues: the authors need to provide all illustrations legend in English, without using Chinese characters
Round 2
Reviewer 1 Report
The authors have edited the manuscript based on my previous review and the changes are, for the most part, satisfactory. Publication is now recommended.
Reviewer 3 Report
The authors did not address my comments. The mechanism of action is unclear. The authors should first perform the experiments to find out the mechanism, then submit the manuscript. They also did not perform any experiments to characterise the material. They provide new images of pure graphene used, but they had to do at least loading efficiency, release profiles and colloid stability of insecticide-loaded graphene. What is the point in showing TEM/SEM, FTIR, Raman of the pure graphene? This is the textbook information. The authors were clearly told to characterise the new material they obtained using graphene, not the raw graphene. Therefore, the paper has to be rejected
